# Carbon Footprint of Away-From-Home Food Consumption in Brazilian Diet

**DOI:** 10.3390/ijerph192416708

**Published:** 2022-12-13

**Authors:** Ilana Nogueira Bezerra, Sara Maria Moreira Lima Verde, Bruno de Sousa Almeida, Clarisse Vasconcelos de Azevedo

**Affiliations:** 1Postgraduate Program in Nutrition and Health, Ceara State University, Av. Dr. Silas Munguba, 1700, Fortaleza 60714-903, Brazil; 2Postgraduate Program in Public Health, Ceara State University, Av. Dr. Silas Munguba, 1700, Fortaleza 60714-903, Brazil

**Keywords:** street food, carbon footprint, food habits, food services, sustainable development indicators

## Abstract

Environmentally unsustainable diets are often characterized by being high in calories, processed foods, and red meats, characteristics related to away-from-home food (AFHF). The aim of this study is to evaluate if AFHF consumption is related to environmental sustainability. Data of 20,780 adults from 24 h recalls collected in the 2017–2018 Brazilian National Dietary Survey (NDS) were used to estimate carbon footprint coefficients. The mean carbon footprint was estimated among individuals who consumed AFHF and non-consumers. Linear regression models were used to evaluate differences between away-from-home eating and the carbon footprint of the diet, adjusting for age and income. A total of 41% of Brazilians consumed AFHF during the previous day. The mean carbon footprint from foods consumed away from home represented 18% of the total carbon footprint. AFHF was positively associated with increased total carbon footprint (β: 204.1; *p*-value: 0.0145). In conclusion, the consumption of foods away from home in urban areas of Brazil was associated with atmospheric greenhouse gas emissions independently of age and income. Away-from-home food consumption should be considered to reinforce the influence of diet on individual and planet health.

## 1. Introduction

Consuming food away from home is an enduring aspect of the Brazilian diet. In 2017–2018, one-third of the expenditures on food were for away-from-home foods (AFHF) [1]. It is well-known that AFHF consumption has a substantial impact on daily caloric intake due to the high content of total fat and saturated fat in foods prepared away from home [2]; however, little attention has been given to the contribution of away-from-home food to environmental sustainability. 

Food is an important link between human health and environmental sustainability [3], and food preferences, including choices of what and where to consume food, are important factors in this relationship [4]. Sustainable healthy diets are dietary patterns that promote all dimensions of individuals’ health and well-being; have low environmental pressure and impact; are accessible, affordable, safe, and equitable; and are culturally acceptable [5,6].

Foods that are frequently consumed away from home are frequently highly processed, high in calories, and include added sugars and saturated fats. Diets based on this type of food have high environmental impact, do not promote the health and well-being of individuals, and go against the principle of sustainable healthy diets [7,8]. In Brazil, a temporal analysis of household food availability showed that environmental effects have increased over the last three decades (carbon footprint increased by 21%, water footprint increased by 22%, and ecological footprint by 17%), along with increased consumption of ultraprocessed foods [9].

The growing number of scientific articles about the environmental impacts of food can be seen as one of the collective efforts needed to contain the progressive environmental degradation. Along with the food and beverage industry, consumers can play an important role in making informed choices that benefit the environment and their own health [10,11,12].

Many studies have examined various attributes of AFHF, such as nutrition, economy, safety, and social issues. However, environmental sustainability is still in its early stage of research and has received insufficient attention from governments [13,14,15,16]. Our hypothesis is that AFHF consumption increases the carbon footprint of the diet. To test this hypothesis, the aim of this study was to evaluate if AFHF consumption is related to environmental sustainability, using Brazilian nationally representative data of food intake.

## 2. Materials and Methods

### 2.1. Study Design and Data Source

This study was based on data from the most recent available national-level nutrition survey conducted by the Brazilian Institute of Geography and Statistics (IBGE), the 2017–2018 National Dietary Survey (NDS). NDS was carried out along with the 2017–2018 Household Budget Survey (HBS). Details of the survey methodology have been published previously [17]. Briefly, the target population was Brazilian households which were selected through an Integrated Household Survey System. This system adopted a two-stage cluster sampling plan, corresponding to a “master sample”, common to all IBGE household surveys. The master sample comprises census tracts, which are the primary sampling units (PSU) selected by systematic sampling with probabilities proportional to size. PSU were stratified by geographic location (urban/rural and five Brazilian geographic areas) and economic level. In the second stage, households were selected by simple random sampling. For HBS, 55,000 households were included, and twenty-five percent of these households were selected to participate in NDS (*n* = 20,112 households). The NDS included all residents over 10 years of age in the selected households, totaling 46,164 individuals. For this paper, only adults residing in urban areas were included (n = 22,780).

Respondents were asked to recall everything they ate and drank, from midnight to midnight during the previous 24 h and to include detailed descriptions and the amounts consumed, based on the multiple-pass method. IBGE processed the data in a specific program. Data collection was carried out over 12 months, uniformly across the strata, ensuring representativeness in the four quarters of the year.

Respondents were asked to report the location where each food was consumed. If the location was anywhere other than “home or brought from home”, then respondents selected a location from a list of pre-coded response categories: school, per-kilo restaurant, fast food restaurants, full-service restaurants, street food, and other places. For this analysis, respondents were considered to have consumed AFHF if they reported at least one item away from home on the previous day.

The amount of food consumed in grams or milliliters was estimated using the tables of measures referred for foods consumed in Brazil, and the nutritional composition was estimated from the nutritional composition table generated for the 2017–2018 survey [1].

Carbon footprint coefficients to quantify the atmospheric emissions of greenhouse gases were estimated based on the publication “Footprints of food and culinary preparations consumed in Brazil” [18] and similar coefficients were used in the case of foods that did not have available estimates. Coefficients took into account all the ingredients included in the preparation, conversion factors, and cooking indices and are expressed in grams of carbon dioxide equivalent per person per day (gCO_2_e/person/day).

### 2.2. Statistical Analysis

Descriptive statistics were used to calculate the proportion of Brazilian adults who consumed AFHF the day before the interview, as well as mean carbon footprint and mean energy intake, on that day, according to place of consumption (at home or away from home). Only data from the first dietary recall were used for this study. Estimates were generated for the overall population, by Brazilian regions, income groups, and age (20–40 and 40–60 years old).

The consumption of food away from home was evaluated as a dichotomous variable (yes/no), identifying whether individuals consumed at least one item away from home in the day before the interview. The mean carbon footprint was estimated among individuals who consumed away-from-home foods and among non-consumers. The mean of carbon footprint was adjusted by total energy intake, using the residual method, and was estimated according to age groups, Brazilian regions, and income levels. Energy intake and carbon footprint were log-transformed to better approximate a normal distribution, and the exponential of the adjusted value was calculated to facilitate the interpretation.

Linear regression models were used to evaluate the relationship between away-from-home eating (independent variable) and the carbon footprint of the diet (dependent variable). Models were developed stratified by each sociodemographic variable (age, income, and Brazilian regions), and the differences of mean were tested through least square mean statement.

The contribution of each place of food consumed away from home (school, per-kilo restaurants, fast food restaurants, full-service restaurants, street food, and other places) to the total carbon footprint from away-from-home food consumption was estimated using the ratio of means method.

All statistical analyses were weighted and performed using survey procedures from SAS release 9.4 (2020, Statistical Analysis Software (SAS Institute Inc.), Cary, NC, USA) to take into account the sample design effect. Weight factors were corrected for non-response, thus allowing population-representative estimates.

## 3. Results

Overall, almost one in two Brazilian adults from urban areas (41.2%) had consumed AFHF during the previous day. The frequency of eating out varied substantially by income and Brazilian regions: Midwest showed the highest frequency of eating out, and away-from-home food intake increased with income. Young adults presented higher frequencies of AFHF consumption (43.9%) (Table 1).

Total mean energy intake was 1770 kcal (95% confidence interval—95%CI: 1750–1790) and away-from-home food consumption contributed to 15% of total energy intake. The mean carbon footprint from foods consumed away from home was 926.3 (Standard Error—SE: 27.9 gCO_2_e/person/day), averaging 17.7% of the total carbon footprint (Figure 1).

Individuals who reported consuming foods away from home presented, on average, a higher carbon footprint than individuals who did not eat away from home (4983 vs. 4725 gCO_2_e/person/day); *p*-value: <0.0022).

The Midwest presented the highest carbon footprint compared to other regions for both consumers and non-consumers of AFHF; however, the difference in the mean carbon footprint between consumers and non-consumers was significant only in the Northeast region. This difference was also higher among young individuals than older adults (Table 2).

Even after controlling for age and income, away-from-home eating was positively associated with increased total carbon footprint (β: 204.1; *p*-value: 0.0145).

Considering only away-from-home food consumption, the place that contributed the most to the production of carbon footprint was other places, followed by self-service and full-service restaurants (Figure 2).

## 4. Discussion

Our study hypothesized that among adults living in urban areas of Brazil, eating away from home would increase the diet’s carbon footprint. In this large sample of 22,780 Brazilians, eating away from home was associated with atmospheric greenhouse gas emissions independent of age and income.

AFHF consumption is well-known as a factor that can increase total energy, total and saturated fats, sugar, and sodium intake [19,20,21]. A high frequency of eating out is also related to poorer diet quality. In general, individuals who consume food away from home present lower intakes of fiber, dairy, fruit, vegetables, and micronutrients than non-consumers [2]. Beyond individual health, away-from-home food has many other implications for the environment that should not be overlooked.

In our study, the carbon footprint of individuals who consume food away from home is greater than among individuals who do not consume food away from home. Our findings reinforce a household survey in 12 Chinese provinces, in which eating out of the home leads to higher climate loads (3505 gCO_2_e/person/day vs. 3191 gCO_2_e/person/day) than eating at home and an increased carbon footprint, from 3293 gCO_2_e/person/day to 3513 gCO_2_e/person/day, over a 7-year period [22].

In our study, AFHF consumption was associated with region of residence (Southeast), age (20–40 years) and income (≥6 minimum wages). In a study that quantified the environmental impacts of food in 37 nations, greenhouse gas emissions significantly increased with income (1.1 kgCO_2_e/person/day in upper-middle-income nations and 2.4 kgCO_2_e/person/day among high-income nations). Furthermore, animal products (meat, fish, and dairy products) account for 22%, 65%, and 70% of emissions in the diets of low-middle-income, upper-middle-income, and high-income nations, respectively. Brazil had different results compared to the nations in its group, with 200% emissions greater than the average of its income groups and with a high contribution related to meat consumption [23].

A recent meta-analysis with 136 articles that evaluated diets, their components, and the environmental impact of their consumption, highlighted that only five studies have focused their analyses on meals (homemade, semi-prepared, ready-to-eat, school snacks, specific preparations) and their impact on the environment [24]. Our findings contribute to clarifying the state of the art about sustainable diets since we bring a perspective focusing on eating away from home and its influence on the carbon footprint. Thus, our findings are in line with the Sustainable Development Goals (SDG) that recommend developing national public policies directed towards primary care integrated with family planning and education in healthy and sustainable diets, as recommended by the EAT-Lancet Commission on Food, Planet, Health [25].

On the other hand, eating homemade food does not necessarily guarantee a healthy diet. From 1987–1988 to 2017–2018, there was an increase of 21% in the carbon footprint from household food availability in Brazil [9]. The carbon footprint is one of the pillars for evaluating food sustainability, being an important instrument to evaluate the potential impact of production processes on the environment. It measures the total amount of greenhouse gas emissions (CO_2_, CH_4_, N_2_O, HFC, PFC, SF_6_, among others) that are directly or indirectly caused by an activity or are accumulated over the course of a product life cycle [18]. Brazil is one of the most populous and productive countries in the world and, as such, is responsible for a large proportion of greenhouse gas emissions, water use, and land cover [9].

The increasing amount of carbon footprint in Brazil might be a result of the 183% increase in the consumption of ultraprocessed foods, which contributed to a 245% increase in greenhouse gas emissions (from 110 gCO_2_e per 1000 kcal to 380 gCO_2_e per 1000 kcal) in the studied period [9]. This is particularly important when we evaluate AFHF intake once ultraprocessed foods (such as fried and baked snacks, soft drinks, pizzas, sweets, and sandwiches) presented the greatest contribution from out-of-home food consumption [8]. The importance of this habit is reinforced by our findings that 18% of the carbon footprint of the Brazilian diet comes from AFHF consumption.

The consumption of ultraprocessed foods in other countries has also shown a direct impact on the environment. In Australia, ultraprocessed products account for 40% of total dietary energy and contribute to more than one-third of all food-related factors that impact the environment, with 35% in relation to land and water use, 39% of energy, and 33% of CO_2_ equivalents [26]. The same study estimated that the consumption of food with empty calories will lead, by 2050, to almost double the per capita emissions of greenhouse gases.

Foods high in salt and sugar contributed approximately 22–23% to greenhouse gas emissions in France [27]. In England, fatty and sugary foods, both in school lunches and in ready-to-eat packaged preparations, represented approximately 8.5% of all greenhouse gases [28]. This high contribution of ultraprocessed products to greater emission of greenhouse gases was also observed in other parts of the United Kingdom [29] and in Japan [30].

Contemporary food has become unsustainable [31]. In this sense, the application of high levels of processing with the use of conservation techniques can represent a high risk for healthy and sustainable food, especially when a large part of the nutrients of the food in its original composition is lost, and fats, sodium, sugars, additives and preservatives are added [32].

Eating away from home is a concern because it is characterized by the consumption of ultraprocessed foods. A recent review on the environmental impact of AFHF found a wide variation of greenhouse gas emissions, depending on the place of food preparation: school canteen meals varied from 0.134 kgCO_2_e/meal to 13.2 kgCO_2_e/meal and other catering services from 0.60 kgCO_2_e/meal to 9.6 kgCO_2_e/meal. Although results are difficult to compare, meat products and food processing are the main factors in the total greenhouse gas emissions for AFHF [33].

An important issue related to the place where food is prepared and/or consumed is that its effect on the environment might vary depending on sociodemographic characteristics. Our analysis shows that eating away from home increases the carbon footprint of the diet depending on the individual’s region of residence (8% for Northeast region). The statistically significant difference between AFHF consumers and non-consumers found only in the Northeast regions reflects the possible greater regional disparity in this region, since consumption of food away from home increases with income [8]. Individuals with greater purchasing power when eating away from home have greater accessibility to different types of food and more expensive foods, which generally involve meat-based preparations [34]. In a Chinese study, AFHF consumption generated more carbon footprint (53%) from animal food products than at-home food consumption (1696 gCO_2_e/person/day vs. 1160 gCO_2_e/person/day, respectively) [22].

One limitation of our study is the definition of AFHF. In Brazilian NDS, if food was consumed at home or brought from home to be consumed in other places it was considered at-home consumption [17]. In addition, foods ordered by food delivery systems to be consumed at home are considered at-home foods. However, the impact of foods bought and consumed directly in the restaurant is different when food is carried out or ordered through online digital platforms. The fact that food comes in a package and needs additional transportation to reach individuals’ households increases the production of waste and air pollution, thus they have different impacts on the environment. According to the 2017–2018 Household Budget Survey, the relative share of ready-to-eat meals in total calories, determined by household food purchases, almost doubled between 2008 and 2018, showing that the influence of foods prepared away from home, but consumed at home, might have major effects on food sustainability.

Although we do not have details about how people had access to their food, we evaluated different places where foods were prepared and consumed. School meals represented the lowest contribution to greenhouse gas emissions. Similar results were found in Finland, where school lunches also had the least environmental impact compared to homemade and ready-to-eat lunches [35]. The category of “other places”, which includes places of consumption different from restaurants (full service, self-service), fast food restaurants or bars, street food, and school cafeterias, showed the highest contribution to carbon footprint consumed from AFHF. We do not have details about the type of place that was considered in “other places”; however, they might include places that do not have as their main purpose the sale of food, such as gas stations, convenience stores, drugstores, newsstands, hospitals, Internet cafes, public or private companies, theaters, movies, or bookstores. As these places do not sell food as their main commercial activity, it is probable that the type of foods most sold there is ultraprocessed products [34].

Healthy and sustainable diets are those with adequate calories, plant-based foods, low amounts of animal foods, polyunsaturated fat rather than saturated fat, and low amounts of refined carbohydrates, ultraprocessed foods, and added sugar [8,36,37]. Changes in current diets leading to healthier and more sustainable choices can substantially benefit human health, with a reduction of almost 24% of deaths worldwide [36]. These changes are necessary both at home and away from home, and it goes beyond individual food choices.

Promoting nutritious and sustainable eating habits requires clearer and more explicit guidance on food choices, amounts, and substitutions that need to be adopted. For example, the lack of quantification in terms of number of servings and portion sizes (especially for products of animal origin) makes it difficult for individuals to adopt them, especially when choosing foods outside the home [38,39]. Effective decarbonization will require a major shift not just in food preferences, but also a reformulation of the recommendations that support this transition [39]. Limiting AFHF consumption is not a simple choice to set in our daily routine; there is a need for public policies that encourage healthier food choices and subsidize more sustainable food production.

Our study points out the necessity to evaluate places where foods are produced and consumed and their influence on the food environment. A positive aspect of our results is that we used a representative sample of Brazilian adults to examine this relationship and reinforce the need for public health strategies for the AFHF sector.

## 5. Conclusions

Our results showed that consuming food away from home is related to environmental sustainability. AFHF consumption in urban areas of Brazil contributed to 17.7% of the total carbon footprint of adults’ diets and was associated with atmospheric greenhouse gas emissions independent of age and income. This relationship varied depending on sociodemographic characteristics. Therefore, away-from-home food consumption has an important role not only for individuals’ health but also for planet health and should be considered to reinforce the influence of diet.

## Figures and Tables

**Figure 1 ijerph-19-16708-f001:**
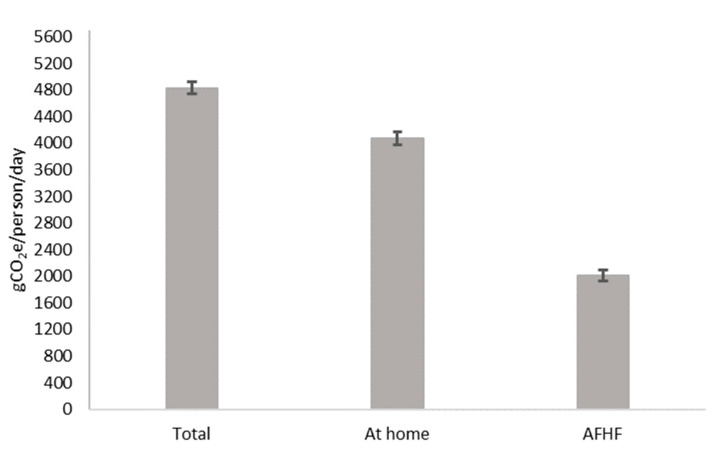
Mean and 95% confidence interval of energy-adjusted carbon footprint (gCO_2_e/person/day) of food consumption according to place of consumption. Brazilian adults in urban areas, 2017–2018 (n = 22,780).

**Figure 2 ijerph-19-16708-f002:**
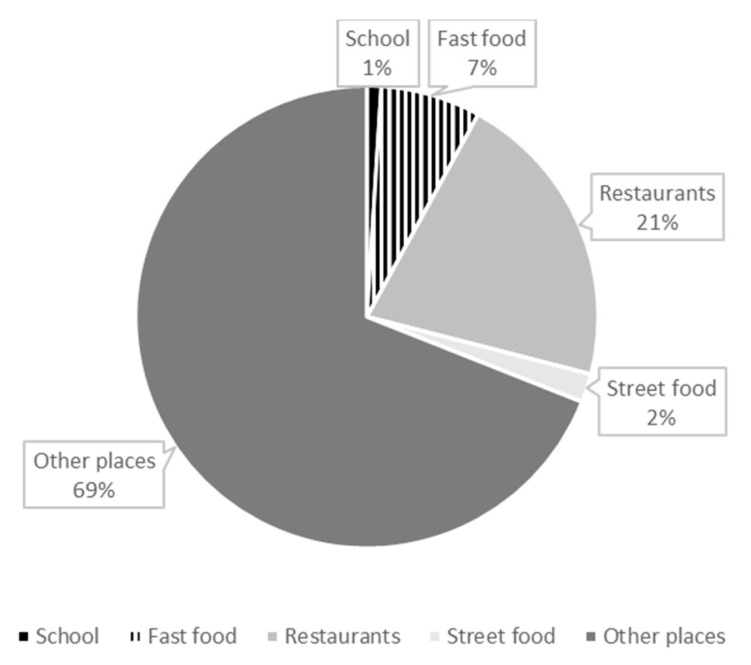
Contribution of place of food consumption to the total carbon footprint from away-from-home food consumption. Brazilian adults in urban areas, 2017–2018 (n = 22,780).

**Table 1 ijerph-19-16708-t001:** Sample distribution and frequency of away-from-home food (AFHF) consumption, according to sociodemographic characteristics. Brazilian adults in urban areas, 2017–2018 (n = 22,780).

Sociodemographic Characteristics	Sample Distribution	AFHF Consumption
Region		
North	7.5 (7.0–8.0)	34.0 (30.2–37.9)
Northeast	23.2 (22.4–23.9)	39.7 (37.9–41.4)
Southeast	46.2 (45.1–47.4)	40.4 (38.2–42.7)
South	14.8 (14.0–15.5)	44.3 (41.3–47.3)
Midwest	8.4 (7.9–8.9)	51.0 (47.8–54.3) ^a^
Age		
20–40 years old	53.5 (52.5–54.6)	43.9 (42.2–45.6)
40–60 years old	46.5 (45.4–47.5)	38.1 (36.6–39.7) ^a^
Per capita household income		
Up to 2 minimal wages	36.9 (35.5–38.3)	30.6 (29.9–32.3)
2–4 minimal wages	45.9 (8.3–10.3)	45.1 (43.3–46.9)
4–6 minimal wages	9.3 (8.3–10.3)	51.9 (47.9–55.9)
≥6 minimal wages	7.9 (6.9–9.0)	55.7 (49.5–61.8) ^a^

^a^*p* < 0.05 for X² test and *p* for linear trend < 0.05 in the case of ordinal variables.

**Table 2 ijerph-19-16708-t002:** Mean energy-adjusted carbon footprint (gCO_2_e/person/day) according to the consumption of away-from-home food (AFHF) and sociodemographic characteristics. Brazilian adults in urban areas, 2017–2018 (n = 22,780).

Sociodemographic Characteristics	Non-Consumers	AFHF Consumers	Difference	*p*-Value
Region				
North	5126 (4824–5427)	5573 (5156–59916)	−447	0.0783
Northeast	4344 (4214–4475)	4682 (4545–4818)	−338	0.0003
Southeast	4646 (4400–4892)	4868 (4678–5058)	−222	0.1486
South	4812 (4502–5123)	4773 (4529–5017)	39	0.8405
Midwest	5896 (5527–6263)	6105 (5839–6372)	−209	0.3554
Age				
20–40 years old	4700 (4524–4875)	5008 (4871–5145)	−308	<0.0055
40–60 years old	4751 (4603–4898)	4951 (4798–5103)	−200	0.0597
Per capita household income				
Up to 2 minimal wages	4550 (4322–4777)	4653 (4476–4831)	−103	0.4623
2–4 minimal wages	4834 (4646–5023)	5068 (4922–5216)	−234	0.0518
4–6 minimal wages	5033 (4643–5424)	5222 (4824–5620)	−189	0.4706
≥6 minimal wages	4824 (4358–5291)	5164 (4828–5501)	−340	0.2301

## Data Availability

Publicly available datasets were analyzed in this study. These data can be found here: https://www.ibge.gov.br/en/statistics/social/population/25610-pof-2017-2018-pof-en.html?=&t=microdados (accessed 10 October 2020).

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
