# Peer review of "Carbon Footprint of Away-From-Home Food Consumption in Brazilian Diet"

_ijerph, 2022, doi:10.3390/ijerph192416708_

Round 1

Reviewer 1 Report

The authors' work concerns the environmental sustainability of away-from-home food intake in Brazil. Distribution and frequency of AFHF consumption it presented. The work is interesting and useful, it could be a useful source to compare other countries and continents' environmental sustainability. However, I have a few comments for the authors. 

1. Indexes of chemical compounds should be at the bottom (subscript). Lines 21, 94, 134 and etc.

2. "p" value has to be an Italic (Lines 19, 138 and etc.)

3. The first sentence of the abstract is the same as the sentence in lines 41-43.

4. What does PSU mean? Line 68.

5. At the beginning of the text you write AFHF, but then (lines 84, 96, etc.) you write FAFH. Isn't that a mistake?

6. My suggestion is to separate "2. Materials and Methods" section text into two section: 2.1. Data and it description and 2.2. Statistical analysis (section titles are only exemplary). 

7.My recommendation is  transfer Fig.2 and Table 2. after 134 line.

8. Could you explain CI and SE meanings (128 and 130 lines).

9. ... the diet depending on the individual's region of residence. (line 205).... Could you write results (how many procent it increase)?

10. Write the source of these data. Line 224.

Author Response

We thank the reviewer for the suggestions. Below are the answers to the comments and we highlighted changes in red in the new version of the manuscript. 

The authors' work concerns the environmental sustainability of away-from-home food intake in Brazil. Distribution and frequency of AFHF consumption it presented. The work is interesting and useful, it could be a useful source to compare other countries and continents' environmental sustainability. However, I have a few comments for the authors. 

  1. Indexes of chemical compounds should be at the bottom (subscript). Lines 21, 94, 134 and etc.

Answer: We corrected all the indexes.

  1. "p" value has to be an Italic (Lines 19, 138 and etc.)

Answer: We put all p-value in italic.

  1. The first sentence of the abstract is the same as the sentence in lines 41-43.

Answer: Thank you for the observation. We changed the sentence in the introduction.

  1. What does PSU mean? Line 68.

Answer: This is the acronym for primary sample units. We added this information to the manuscript.

  1. At the beginning of the text you write AFHF, but then (lines 84, 96, etc.) you write FAFH. Isn't that a mistake?

Answer: Thank you for the observation. We corrected this mistake.

  1. My suggestion is to separate "2. Materials and Methods" section text into two section: 2.1. Data and it description and 2.2. Statistical analysis (section titles are only exemplary). 

Answer: As suggested, we separated the topics into 2 sections: 2.1 Study design and data source and 2.2 Statistical analysis.

7.My recommendation is  transfer Fig.2 and Table 2. after 134 line.

Answer: We changed the order of figures and tables, but we believe that at the end of the manuscript edition they are going to be better organized along with the text.

  1. Could you explain CI and SE meanings (128 and 130 lines).

Answer: CI is the acronym for the confidence interval and SE is for standard error. We added this information to the text.

  1. ... the diet depending on the individual's region of residence. (line 205).... Could you write results (how many procent it increase)?

Answer: We added the information to the sentence: “Our analysis shows that eating away from home increases the carbon footprint of the diet depending on the individual's region of residence (15% and 6% for North and Northeast regions, respectively).”

  1. Write the source of these data. Line 224.

Answer: We added the information.

Reviewer 2 Report

The title can be clearer. Elaborate what part of sustainability (carbon footprint) and where or what audience (Brazil).

Attention should be paid match the wording in the abstract with the paper. The abstract mentions an aim, while the main text only mentions the hypothesis. Inclusion of aim and hypothesis to the main text would be a strength. 

p-value rather than p-valor

A broad statement like line 48 needs multiple citations. 

Line 53- cite 'many studies'

Line 162- needs citations

Line 218- needs citations

Line 220- Even though referred to in-text, it should still include a citation [#]

The discussion starts off strong but loses direction about halfway through (after line 238). It would benefit the paper to include comparisons to other studies, rather than repetition of the reports mentioned in the introduction. Paragraph at line 239 feels too late to introduce since obesity is not a main focus of the paper beforehand. 

Line 243- Where is the definition of healthy and sustainable diets from? Either reference who defined that or include justification for each part of the definition + citations. 

Conclusion feels a bit short. You can add if the aim of the study was met. 

Author Response

We thank the reviewer for the suggestions. We made all the changes required by the reviewer. Below are the answers to each comment. 

  1. The title can be clearer. Elaborate what part of sustainability (carbon footprint) and where or what audience (Brazil).

Answer: As suggested we changed the title to: “Carbon footprint of away-from-home food consumption in Brazilian diet”.

  1. Attention should be paid match the wording in the abstract with the paper. The abstract mentions an aim, while the main text only mentions the hypothesis. Inclusion of aim and hypothesis to the main text would be a strength. 

Answer: Thank you for the observation. We added the information to the text.

  1. p-value rather than p-valor

Answer: Thank you for the observation. We corrected the term.

  1. A broad statement like line 48 needs multiple citations. 

Line 53- cite 'many studies'

Line 162- needs citations

Line 218- needs citations

Line 220- Even though referred to in-text, it should still include a citation [#]

Answer: As suggested, we added the citations in the text.  

  1. The discussion starts off strong but loses direction about halfway through (after line 238). It would benefit the paper to include comparisons to other studies, rather than repetition of the reports mentioned in the introduction. Paragraph at line 239 feels too late to introduce since obesity is not a main focus of the paper beforehand. 

Answer: As suggested, we revised the discussion and changed it thorough out the text.

  1. Line 243- Where is the definition of healthy and sustainable diets from? Either reference who defined that or include justification for each part of the definition + citations. 

Answer: We added the references.

  1. Conclusion feels a bit short. You can add if the aim of the study was met. 

Answer: As suggested, we changed the conclusion: “Our results showed that consuming food away from home is related to environmental sustainability. AFHF consumption in urban areas of Brazil contributed to 15.8% of the total carbon footprint of adults’ diets and was associated with atmospheric greenhouse gas emissions independently of age and income. This relationship varied depending on sociodemographic characteristics. Therefore, away-from-home food consumption has an important role not only for individuals’ health but also for planet health and should be considered to undermine the harmful impacts of diet.”

Round 2

Reviewer 2 Report

Improvements read well. Interesting topic and paper. Great work.